# Chemical Composition and Biological Activity of Argentinian Propolis of Four Species of Stingless Bees

**DOI:** 10.3390/molecules27227686

**Published:** 2022-11-09

**Authors:** Valery A. Isidorov, Jolanta Maslowiecka, Lukasz Szoka, Naldo Pellizzer, Dora Miranda, Ewa Olchowik-Grabarek, Monika Zambrzycka, Izabela Swiecicka

**Affiliations:** 1Institute of Forest Sciences, Bialystok Technical University, 15-351 Bialystok, Poland; 2Department of Medicinal Chemistry, Medical University of Bialystok, 15-222 Bialystok, Poland; 3Facultad de Ciencias Forestales, Universidad Nacional de Misiones, Bertoni 124, Eldorado 3380, Misiones, Argentina; 4Department of Microbiology, Biological Faculty, University of Bialystok, 15-328 Bialystok, Poland

**Keywords:** stingless bees, propolis, chemical composition, plant sources, anticancer activity, antimicrobial activity

## Abstract

The chemical composition of propolis of four species of stingless bees (SLBs) from Argentina was determined, and its antibacterial and anticancer activity was evaluated on selected types of microbes and cancer cell lines. Volatile secretions of all propolis samples are formed by 174 C_2_–C_15_ organic compounds, mainly mono- and sesquiterpenes and their derivatives. The chromatograms of ether extracts showed 287 peaks, of which 210 were identified. The most representative groups in the extracts of various propolis samples were diterpenoids (mainly resin acids), triterpenoids and phenolic compounds: long-chain alkenyl phenols, resorcinols and salicylates. The composition of both volatile and extractive compounds turned out to be species-specific; however, in both cases, the pairwise similarity of the propolis of *Scaptotrigona postica* and *Tetragonisca fiebrigi* versus that of *Tetragona clavipes* and *Melipona quadrifasciata quadrifasciata* was observed, which indicated the similarity of the preferences of the respective species when choosing plant sources of resin. The composition of the studied extracts completely lacked flavonoids and phenolcarboxylic acids, which are usually associated with the biological activity and medicinal properties of propolis. However, tests on selected microbial species and cancer cell lines showed such activity. All propolis samples tested against *Paenibacillus larvae*, two species of *Bacillus* and *E. coli* showed biofilm inhibition unrelated to the inhibition of bacterial growth, leading to a decrease in their pathogenicity. Testing the anticancer activity of ether extracts using five types of cell cultures showed that all four types of propolis studied inhibit the growth of cancer cells in a dose- and time-dependent manner. Propolis harvested by *T. clavipes* demonstrated the highest cytotoxicity on all tested cell lines.

## 1. Introduction

Meliponines, also called stingless bees (SLBs), live in subtropical and tropical regions, which differ from other geographical areas in their huge floristic diversity. As pollinators, SLBs are involved in maintaining and preserving the diversity of flowering plants growing there, while a rich plant life offers the bees a rich choice of various kinds of materials that serve as food and are used for other purposes, such as building a nest and maintaining hygienic conditions in it, necessary for breeding healthy offspring and ensuring the safety of accumulated food reserves. A particularly important role in this is played by mixtures of plant resins with bee wax called propolis (if soil or clay is added to them, the mixture is called geopropolis). It is well known that the creation of stocks of propolis is the most important evolutionary method developed by bees to prevent the occurrence of microbial infections that are detrimental to organisms living in close quarters [1,2]. In the life of SLBs, these kinds of mixtures play a particularly important role, since they are used more widely, in particular (unlike the honeybee, *Apis mellifera*) for building a nest and as pots for rearing offspring and for storing supplies [3]. Therefore, most species of stingless bees collect more resin for making propolis than honeybees [4].

The beneficial properties of propolis have been known for a long time: the roots of its use by humans for medicinal purposes go back to the times of Ancient Egypt and Greece in the Old World and to the time of the Incas and Aztecs in the New World. The protective (antimicrobial) and medicinal properties of propolis, useful both for the bees themselves and for humans, are associated with the presence of biologically active substances of plant origin in it: the wider the palette of these compounds in the resins provided by vegetation to bees, the higher the medicinal value of the propolis prepared by them [5]. Therefore, it is not surprising that many researchers note a wider spectrum of action and a higher biological activity of SLB propolis, which is characterized by a huge variety of plant-derived compounds with valuable biological properties and with different mechanisms of action [6,7,8]. A number of studies have shown the antimicrobial and antiviral activities of SLB propolis [9,10,11,12,13,14,15,16], its cytotoxic effect on some cancer cell lines [5,17,18,19,20] and its antioxidant and anti-inflammatory actions [9,13,14,21,22], as well as the prevention of a number of metabolic diseases [23].

Tropical areas are home to many resiniferous plant species [24], as well as many species of eusocial SLBs that differ in many ways, including their preference for certain plant sources of resins, and this leads to great diversity in the types of propolis prepared by them and is a problem for researchers. Probably, to date, the most chemical and medicinal information has been obtained about SLB propolis from the tropical regions of South America, primarily Brazil [6,9,14,15,17,25,26,27,28,29,30,31,32]. However, it is far from complete and concerns the propolis of a few neotropical SLBs. For example, out of more than 242 species of SLBs described in Brazil in 2014 [33], the propolis and geopropolis composition of only a few species (*Frieseomelitta longipes, F. silvestrii, F. silvestrii languida, F. varia*, *Melipona quadrifasciata. M. quadrifasciata anthidioides. M. fasciculata, M. interrupta*, *M. orbignyi*, *M. seminigra, M. scutellaris, M. subnitida, Tetragonisca angustula, T. fiebrigi, Scaptotrigona aff. postica, S. bipunctata, S. depili*, etc.) has been studied. There is even less information about Argentine SLB propolis and geopropolis. The antimicrobial, anti-inflammatory, antitussive and expectorant activities, as well as the cytotoxicity of the propolis of two species of SLBs, *S. jujuyensis* and *T. fiebrigi*, from north-western Argentina have been reported but without detailed information about its chemical composition [11,22].

The aim of this study was to determine the chemical composition of the propolis of four species of SLBs from the north-eastern part of Argentina and to evaluate their anticancer and antimicrobial activities. We tried to characterize the composition of the studied propolis samples as fully as possible and therefore resorted to various options for isolating and identifying the components. A combination of headspace analysis and solid-phase microextraction with gas chromatography–mass spectrometry (HS-SPME/GC-MS) was used to characterize volatile components. To determine the composition of less volatile components, they were extracted with diethyl ether, followed by derivatization and GC-MS analysis.

## 2. Materials and Methods

### 2.1. Chemicals

A silylation agent, bis(trimethylsilyl)trifluoroacetamide (BSTFA), with the addition of 1% of trimethylchlorosilane as well as C_8_–C_40_ *n*-alkane calibration standards were purchased from Sigma-Aldrich (Poznań, Poland). The solvents used for extraction (diethyl ether and methanol) were purchased from POCH (Gliwice, Poland).

### 2.2. Material

In this work, we determined the composition of the volatile and extractive components of propolis of four species of stingless bees, the scientific and local names of which are given in Table 1. Samples 1, 2 and 4 were taken on 25 August 2021 from the educational Meliponario “Paulo Neto-Nogueira” of the Faculty of Forestry Sciences, Eldorado Misiones, Argentina (54°39′51′′ W–26°24′17′′ S). Sample 3 was taken on 24 August 2021 from the “Alto Uruguay” Meliponary in the town of San Vicente, Misiones, Argentina (54°32′03′′ W–26°59′40′′ N).

### 2.3. Headspace/Solid-Phase Microextraction for Volatile Analysis

To investigate the volatile organic compounds (VOCs) of propolis, the analytical procedure previously described by Isidorov et al. [34,35,36], HS-SPME/GC-MS, was used. In these works, it was experimentally established by, for example, determining the VOC of the propolis or buds of different species of birch, where the best results are achieved using divinylbenzene/carboxen/PDMS (DVB/CAR/PDMS) sorption fibre, and the conditions of the experiment were also optimized.

The propolis was cooled to −18 °C and ground. Next, 0.5 g of the powder was placed into a 16 mL headspace vial and immersed into a thermostat at 40 °C. The membrane of the screw cap was pierced by a needle with DVB/CAR/PDMS fibre and exposed to the headspace gas phase. After 50 min of exposure, the fibre was placed for 10 min into the injection port of an HP7890A gas chromatograph with the 5975C VL MSD Triple-Axis Detector (Agilent Technologies, Santa Clara, CA, USA). The apparatus was fitted with an HP-5ms capillary column (30 m × 0.25 mm i.d., 0.25-μm film thickness), with electronic pressure control and a split/splitless injector. The latter was operated at 250 °C in splitless mode. The helium flow rate through the column was 1 mL min^−1^ in constant flow mode. The initial column temperature was 40 °C and rose to 220 °C at a rate of 3 °C min^−1^. The MSD detector acquisition parameters were as follows: the transfer line temperature was 280 °C, the MS source temperature was 230 °C and the MS quad temperature was 150 °C. Electron impact mass spectra were obtained at 70 eV of ionization energy. Detection was performed in full scan mode from 39 to 350 a.m.u. After integration, the fraction of separated components in the total ion current (TIC) was calculated.

To calculate linear-temperature-programmed retention indices (*RI*) of the analytes, SPME fiber was inserted for 2−3 s into the headspace of the vial with a mixture of C_6_−C_18_ *n*-alkanes. Their separation was performed under the above-mentioned conditions.

### 2.4. Extraction of Propolis

An aliquot (1 g) of the powdered propolis was transferred into a flask (50 mL) and extracted by stirring with three 25-mL portions of diethyl ether for 30 min. The combined extracts were filtered through paper filter, and the solvent was evaporated in a fume hood at room temperature.

A portion (5–6 mg) of the dry residue after ether extraction was dissolved in 220 µL pyridine, and 80 µL of BSTFA was added. The mixtures were sealed and heated for 50 min at 60 °C to form trimethylsilyl (TMS) derivatives. GC-MS analysis of the derivatized extracts was performed using the above-mentioned GC-MS apparatus using an HP-5ms capillary column at a helium flow rate of 1 mL min^−1^. An injection of a 1-µL sample was performed using an Agilent 7693A autosampler. The injector worked at a temperature of 300 °C in split (1:20) mode. The initial column temperature was 50 °C, rising to 320 °C, at 3 °C min^−1^; the final temperature was held for 10 min. The ion source and quadrupole temperatures were 230 °C and 150 °C, respectively. Electron ionization mass spectra (EIMS) were obtained at an ionization energy of 70 eV. Detection was performed in full scan mode from 41 to 650 a.m.u.

The hexane solution of C_10_–C_40_ *n*-alkanes was separated under the above-mentioned conditions. Retention indices values were calculated from the results of the separation of this mixture and the solutions of components extracted from propolis.

### 2.5. Component Identification

To identify the components obtained from the extracts, two independent analytical parameters were used: mass spectra and calculated retention indices. The mass spectrometric identification of propolis volatiles was carried out using an automatic system for GC-MS data processing supplied by the NIST 14 library (NIST/EPA/NIH Library of Electron Ionization Mass Spectra), as well as by computer search libraries containing the mass spectra and retention indices from Adams’ [37] and Tkachev’s [38] collections. The retention indices of the components registered in the form of TMS derivatives were compared with those presented in the NIST collection [39] and in the recently published database [40]. The latter contains mass spectra and *RI* values of 1725 TMS derivatives prepared mainly from commercial preparations of flavonoids, other phenolics, terpenoids, aliphatic and aromatic acids, alcohols, carbohydrates and glycosides. The identification was considered reliable if the results of the computer search of the mass spectra library were confirmed by the experimental *RI* values, i.e., if their deviation from the published database values did not exceed ±10 u.i. (the average quantity of inter-laboratorial deviation for non-polar stationary phases). If the result of mass spectrometric identification was not confirmed chromatographically due to the absence of retention index values in the available databases, or if the calculated and literature index values differed by more than 10 u.i., the identification was considered tentative.

### 2.6. Cell Viability Tests

#### 2.6.1. Cell Culture

Tongue squamous cell carcinoma cells (SCC-25), melanoma cells (C32, A375), colorectal adenocarcinoma cells (DLD-1), gastric adenocarcinoma cells (AGS) and normal human skin fibroblasts (CCD25Sk) were purchased from the American Type Culture Collection (ATCC, Manassas, MD, USA). SCC-25 cells were cultured in Dulbecco’s Modified Eagle Medium (DMEM):Nutrient Mixture F-12 (DMEM/F12) supplemented with 10% foetal bovine serum, 1% penicillin/streptomycin (all reagents obtained from Thermo Fisher Scientific, Waltham, MA, USA) and 400 ng mL^−1^ hydrocortisone (Sigma-Aldrich, Saint Louis, MO, USA). The other cell lines were cultured in DMEM (Gibco) supplemented with 10% foetal bovine serum and 1% penicillin/streptomycin. Cell culture was maintained at 37 °C in a humidified atmosphere containing 5% CO_2_.

#### 2.6.2. Cell Viability Assay

Cell viability was evaluated by MTT assay. Cells were seeded in 96-well plates at a density of 1 × 10^4^ cells per well and incubated for 24 h. Next, the cells were treated with various concentrations of propolis extracts (1.5, 3.1, 6.2, 12.5, 25, 50, 100, 200, 400 and 800 μg mL^−1^) for 24 and 48 h. Propolis extracts were dissolved in DMSO before adding them to the cell culture, and 0.2% DMSO served as a control. 3-(4,5-Dimethyl-2-thiazolyl)-2,5-diphenyl-2*H*-tetrazolium bromide (MTT; Sigma-Aldrich, Saint Louis, MO, USA) solution was added to each well, and the cells were incubated at 37 °C. After a 4 h incubation period, MTT was removed and DMSO-Sorensen’s glycine buffer (8:1; *v/v*) was added to each well. The optical density was measured at 570 nm using a microplate reader. The results were expressed as cell viability rates.

#### 2.6.3. Statistical Analysis

The data were from three independent experiments and expressed as means ± SEM. IC_50_ values were calculated using GraphPad Prism software (version 7.04). The results were analysed in GraphPad Prism software using one-way ANOVA followed by Tukey’s test, accepting *p*-values less than 0.05 as significant.

### 2.7. Screening for Antimicrobial Activity

#### 2.7.1. Minimal Inhibitory, Minimal Bactericidal and Minimal Fungicidal Concentrations

The diethyl ether extracts of SLB propolis were tested against microorganisms obtained from the ATCC: *Staphylococcus aureus* ATCC 6538, *Paenibacillus larvae* ATCC 9545, *Bacillus cereus* ATCC 10987, *Bacillus subtilis* ATCC 6633, *Escherichia coli* ATCC 11229, *Pseudomonas aeruginosa* ATCC 19582 and *Candida albicans* ATCC 90029. All the microorganisms kept at −80 °C in the storage medium (LB broth and glycerol in a ratio of 1:1) were inoculated onto nutrient agar (bacteria) or Sabouraud agar (*C. albicans*) and incubated overnight at 37 °C.

The antimicrobial activity of the propolis extracts was assessed by determining the minimal inhibitory concentration (MIC) in accordance with the Clinical and Laboratory Standard Institute (CLSI) protocols [41], as applied previously [35]. In brief, the SLB propolis extracts were dissolved in DMSO at a concentration of 8 mg mL^−1^, filtered with a 0.22-µm-pore-size Rotilabo–Spritzenfilter filter (Carl Roth GmbH and Co, Karlsruhe, Germany) and serially twofold diluted in Mueller–Hinton broth, ranging from 4000 to 0.0002 µg mL^−1^, in a U-shaped 96-well microtitre plate with a final volume of 100 μL. The bacteria were cultured overnight in Mueller–Hinton broth at 37 °C with shaking at 200 rpm and then suspended to a final optical density of 0.2–0.3 at 600 nm wavelength measured with a V-670 spectrophotometer (Jasco Corp., Tokyo, Japan). For the assay, 100 μL of the bacterial suspensions was added to each well in the microtitre plate containing diluted propolis extracts and incubated overnight at 37 °C. To obtain comparable data, all the bacteria were treated under the same conditions. The MIC values were determined as the lowest concentration of the extracts in the wells with no bacterial growth observed visually. All the tests were carried out in quadruplicate, and the results were averaged.

In addition, the minimal bactericidal concentration (MBC) and minimal fungicidal concentration (MFC) of the extracts were assessed. For this purpose, 5 µL of the overnight culture from each well in the microtitre plate with extracts of a concentration equal to and higher than the MIC value were inoculated onto BHI agar with the use of a sterile plastic spreader and incubated overnight at 37 °C. The MBC/MFC values were determined as the lowest concentration of the extracts in the wells with no bacterial growth on the plates observed visually. All the tests were carried out in quadruplicate, and the results were averaged.

All the microbiological media used in the study were supplied by Oxoid Ltd. (Basingstoke, UK). As a positive control, microorganisms cultured in Mueller–Hinton broth and on BHI agar without the propolis extracts were applied. Mueller–Hinton broth supplemented with 10% DMSO was used as the solvent control, while Mueller–Hinton broth with 10% DMSO and extracts was used as the propolis extract control. The MIC and MFC values for *C. albicans* were assessed as mentioned before but with the application of Sabouraud broth and Sabouraud agar instead of Mueller–Hinton broth and BHI agar, respectively.

#### 2.7.2. Biofilm Formation Assay

Biofilm formation was determined for strains of *P. larvae* ATCC 9545, *B. cereus* ATCC 10987, *B. subtilis* ATCC 6633 and *E. coli* ATCC 11229. Bacterial suspension was prepared, as described before (Section 2.7.1). Next, the suspension was incubated with 1/32 MIC, 1/16 MIC, 1/8 MIC, ¼ MIC and ½ MIC of SLB extracts in a 96-well plate at 37 °C for 48 h. After incubation, the planktonic cells were removed. The biofilms were washed three times with sterile water. After drying, 200 μL of 0.1% crystal violet dye was added to each well and incubated for 15 min at room temperature. The biofilms were then rinsed with sterile water to remove the dye. After drying, 200 μL of DMSO was added to every well and incubated for 10 min at 37 °C. Next, absorbance at 570 nm was measured using a SpectraMax M2 microplate reader (Molecular Devices).

The research was performed in four independent experiments. Data are presented as the mean ± SD. The level of significance was analysed using one-way ANOVA, and *p* < 0.05 and below was accepted as statistically significant. Statistical analysis was performed using Origin 8.5.1 software (Microcal Software Inc., Northampton, MA, USA).

## 3. Results and Discussion

### 3.1. Chemical Composition of Volatile Compounds

Probably the least studied aspect of propolis is the composition of its volatile organic compounds (VOCs), since a common approach to extracting them from the raw material is maceration followed by evaporation of the solvent, during which the most volatile components are inevitably lost. A number of studies have used labour-intensive techniques of sorption concentration [42] or steam stripping [43,44] to determine the composition of VOCs. A much more efficient and less time-consuming approach is to use the HS-SPME/GC-MS technique, which combines the stages of concentration and sample preparation, which has proven itself well in the study of VOCs of European [36,45] and Brazilian [46] honeybee propolis.

As a result of the determination of volatile substances by this method, 174 peaks were registered on the chromatograms of all four samples of propolis from SLBs. Typical VOC chromatograms are shown in Figure 1. Each of the chromatograms contains 57 to 105 peaks of organic C_2_–C_15_ compounds of various classes. Registered components are divided into nine groups, which are shown in Table 2 together with the main representatives of each of them. The complete composition of the volatiles, along with some of the analytical parameters used to identify them, is shown in Appendix A.

Although the VOCs of each of the four types of propolis contain representatives of all nine groups, their individual composition is quite specific: only 16 compounds (less than 10%) were common for all samples. Nine of them belonged to terpenes characteristic of plant essential oils (C_10_H_16_ monoterpenes α- and β-pinenes, myrcene, 3-carene and limonene, as well as C_15_H_24_ sesquiterpenes α-copaene, β-caryophyllene, α-humulene and δ-cadinene). Other common components were ethanol, acetic acid and their esters, isopentanol, 2-ethylhexanol and its acetate, and the aromatic hydrocarbons styrene and *p*-cymene. However, one can note the pairwise similarity of the VOC composition of propolis samples 1 and 4, as well as samples 2 and 3. VOCs of propolis 1 and 4 are characterized by a relatively low content of terpene compounds but a high content of C_2_–C_9_ alcohols and their esters, while terpenoids prevail in the secretions of propolis 2 and 3.

### 3.2. Extractive Compounds

Substances extracted with diethyl ether from SLB propolis are mainly relatively low-polarity compounds with one or two functional (carboxyl and/or alcohol) groups. The chromatograms of all four samples showed peaks of 287 compounds, of which 210 were identified based on mass spectrometric and chromatographic information. The most representative groups were formed by diterpenes and triterpenes (60 and 62 compounds, respectively), which also accounted for the largest part of the total ion current (TIC) of the chromatograms. The third-largest group (26 compounds) with a much smaller contribution to the TIC was formed by phenol derivatives, long-chain (C_15_–C_19_) alkyl- and alkenylphenols, resorcinols and salicylates. On the chromatograms of *T. clavipes* propolis, 20 sesquiterpene compounds were registered; however, most of them belonged to minor components, the individual concentration of which did not exceed 0.1% TIC. Typical chromatograms of ether extracts are shown in Figure 2. The group composition of these extracts is given in Table 3, but the full composition is shown in Appendix A.

Of the total identified compounds, only six were present in all samples: diterpenes, abietic, dehydroabietic, isopimaric and 13-*epi*-cupressic acids, and pentacyclic triterpene alcohols, α- and β-amyrins. Thus, the composition of the diethyl ether extracts of the propolis of the four species of SLBs that we are interested in is also highly specific, but as in the case of VOCs, there is a pairwise similarity of its group composition. This similarity is clearly demonstrated by the dendrogram in Figure 3, built according to the data on the group composition of the components given in Table 2 and Table 3. The main constituents of samples 2 and 3 were diterpenoids (mainly diterpene acids), which were contained in propolis 1 and 4 only in small amounts. However, the latter were characterized by a high content of triterpenoids. In addition, phenolic lipids, alkylphenols, alkylresorcinols and alkylsalicylates were found only in them.

It is worth mentioning that a pairwise similarity (samples 1 and 4 vs. 2 and 3) was also observed in the composition of pot honey of the same four SLB species from the same region of Argentina and collected simultaneously with propolis samples 1–4 (our unpublished data). All this together testifies to the peculiarities of the preferences of different species of bees for collecting nectar and plant resins (as well as the coincidence of their collecting preferences). Indeed, the similarity of food preferences of *C. postica* (1) and *T. fiebrigi* (4) bees was shown previously on the basis of palynological analysis of honey prepared by them: plant nectar of the families Fabaceae, Arecaceae and Euphorbiaceae was the main food resource of these bees [47,48]. However, the source of resin for the manufacture of propolis may be plants from other families.

Since the composition of propolis depends on the plants that the bees visit to collect the resin and it contains chemical markers of these plants [34,35,49,50], it is possible to make assumptions about the botanical origin of the studied SLB propolis sample, based on previously obtained information about propolis from other regions and even other bee species.

The most likely plant precursor of propolis samples 2 and 3, the main components of which are diterpene acids, also called resin acids, are the resinous secretions of coniferous trees. The high content of these compounds in propolis is typical for the Mediterranean region [51], whose flora includes various species of cypresses and pines. In the Neotropics of the Western Hemisphere, coniferous trees of the Pinaceae and Araucariaceae families grow, the resinous secretions of which contain diterpene acids, but their individual composition is different. Bankova et al. [52,53] found labdane-type diterpenoids in Brazilian propolis and concluded that their source is most likely local species of *Araucaria*. However, it is impossible to exclude plants of the family Pinaceae from the list of possible sources of diterpenes: Marcucci et al. [54] found dehydroabietic and abietic acids in honeybee propolis from beehives in a natural pine forest in the state of São Paulo (Brazil). Considering the fact that near the places where meliponines are localized in the state of Misiones (Argentina), from which samples 1 and 4 were taken, there are industrial slash pine (*Pinus elliottii*) plantations, and we are inclined to think that these trees serve as a source of diterpenoids in them. This assumption is also supported by the qualitative composition of propolis terpenoids: the ether extracts contained all eight resin acids (abietic, dehydroabietic, isopimaric, levopimaric, neoabietic, palustral, pimaric and sandaracopimaric), and the VOCs contained all eight monoterpenes (α-pinene, β-pinene, camphene, myrcene, 3-carene, limonene, terpinolene and β-phellandrene) characteristic of resins of all pine species [55].

Triterpenoids and lipids with a phenolic core, the main components of samples 1 and 4, have been found in propolis from tropical regions, both from families of honeybees [56,57] and some species of SLBs [15,21,50]. The resins of *Mangifera indica*, belonging to the Anacardiaceae family, have been named as a plant source of these lipids. Reliable evidence in favour of this is the presence in propolis of such triterpenes as mangiferolic, isomangiferolic and mangiferonic acids (Table 3). It cannot be ruled out that other resiniferous representatives of the Anacardiaceae family serve as a source of phenolic lipids found in propolis from South America [29].

Propolis samples from *S. postica* bees from Barra do Corda, Maranhao State and Rio Grande do Norte State (northeast Brazil) have been reported to contain phenolic lipids, but their main components are flavonols, such as quercetin methyl esters and methoxychalcones [21]. As their alleged precursor, the secretions on the tops of the shoots of *Mimosa tenuiflora* are considered. However, these compounds were completely absent in Argentine propolis sample 1 from the same SLB species. A complete discrepancy in the composition of propolis collected by *T. fiebrigi* bees in Brazil and Argentina was also observed: phenylpropenoids absent from Argentinian propolis were found in Brazilian propolis by Campos et al. [9], but triterpenoids and phenolic lipids, the main components of sample 4 (Table 3), were completely absent in Brazilian propolis. It is likely that SLBs do not show strong selectivity in resin collection and use different resources for the production of propolis, provided by local vegetation in the equatorial regions of Brazil and in the subtropics of Argentina.

A feature of the chemical composition of the studied propolis is the absence of flavonoids and phenolcarboxylic or cinnamic acids and their derivatives, which is attributed to the biological activity and medicinal properties of this bee product [58,59]. However, this does not mean that it is deprived of such activity and properties, and this was demonstrated previously [11] using the example of propolis from two SLBs, *T. fiebrigi* and *S. jujuyensis*. Alcoholic extracts of both types of propolis, practically devoid of flavonoids (their content was 0.08%), nevertheless exhibited antimicrobial, antioxidant and antinociceptive activities. It is of interest to further study the medicinal properties of propolis with a ‘non-flavonoid’ composition. For this purpose, we determined the anticancer and antimicrobial activities of extracts from the studied propolis samples.

### 3.3. Anticancer Activity of Ether Extracts

To determine the effect of four propolis ether extracts on the viability of A375, C32, SCC-25, AGS and DLD-1 cells or fibroblasts, the cells were treated with extracts at concentrations of 1.5, 3.1, 6.2, 12.5, 25, 50, 100, 200, 400 and 800 μg mL^−1^ for 24 or 48 h and their viability was assessed using MTT assay (Appendix A). The purpose of the experiment was to establish the dependence of cell survival on the concentration of propolis extract and on the duration of exposure. Based on the dose–response curves, the IC_50_ values were calculated (Table 4).

At the 24 h time point, IC_50_ values of all propolis extracts in A375 cells were above 50 μg mL^−1^. After 48 h of treatment, the cytotoxic effect was stronger and IC_50_ values of extracts 1, 2, 3 and 4 were 29.4 ± 2.2, 16.8 ± 1.5, 43.9 ± 3.6 and 36.1 ± 2.4 μg mL^−1^, respectively. At the 24 h time point, C32 cells were more susceptible to the cytotoxic action of propolis extracts than were A375 cells. Prolongation of the incubation time to 48 h led to a decrease in the IC_50_ value of extract 2 to 17.3 ± 1.3 μg mL^−1^, whereas the viability of C32 cells treated with the other three extracts reduced slightly.

After 24 h of treatment, the viability of SCC-25 cells was highly suppressed by extracts 1, 2 and 4. The cytotoxic action of extract 3 was still high (IC_50_: 42.9 ± 3.5 μg mL^−1^) and similar to that observed in C32 cells. However, prolonged incubation of SCC-25 cells with extracts resulted in a decrease in IC_50_ values only by about 10–20%.

The viability of AGS cells after 24 h of incubation was most strongly inhibited by extract 2 (IC_50_: 38.2 ± 2.4 μg mL^−1^). For the other three extracts, IC_50_ values were slightly higher than 50 μg mL^−1^. At the 48 h time point, IC_50_ values of all propolis extracts decreased significantly and the lowest value (12.6 ± 0.9 μg mL^−1^) was found for extract 2.

Treatment of DLD-1 cells for 24 h with propolis extracts resulted in a relatively slight decrease in cell viability (IC_50_ > 50 μg mL^−1^). However, incubation of cells for 48 h significantly suppressed the proliferation of DLD-1 cells, and the IC_50_ values of extracts 1, 2, 3 and 4 were 31.9 ± 2.0, 20.2 ± 1.5, 32.9 ± 2.4 and 44.3 ± 3.6 μg mL^−1^, respectively. At the 24 h time point, the IC_50_ values of propolis extracts in normal skin fibroblasts were above 50 μg mL^−1^. At the 48 h time point, the IC_50_ value of extract 3 was still above 50 μg mL^−1^, while in the case of the remaining three extracts, it decreased to 34–43 μg mL^−1^.

Thus, the data from the performed experiments indicate that all tested propolis extracts inhibit the growth of cancer cells in both a dose- and a time-dependent manner. For all cancer cell lines tested, propolis extract 2 collected by *T. clavipes* showed the highest cytotoxicity.

It is noteworthy that propolis extract 3 is characterized by higher IC_50_ values than those of extract 2, which is close to it in composition: the main components of both are diterpene acids. At present, it is difficult to explain this discrepancy. However, it can be assumed that in the case of propolis extract 2, synergy with triterpenes is manifested, the relative total content of which in it is much higher: 12.54% versus 0.39% in extract 3. In particular, the content of α- and β-amyrins, triterpene alcohols with well-documented anticancer activity, in the former is much higher. However, this assumption about the presence of synergy needs to be tested.

### 3.4. Antibacterial Activity

To evaluate the antibacterial activity of propolis ether extracts, we determined the MIC, as well as the MBC and MFC. The test results are shown in Table 5 along with our earlier data on MIC values for samples of European honeybee propolis, mainly ‘poplar’ and mixed ‘birch’ and ‘aspen’ types. As can be seen from the data presented, all tested extracts inhibited the growth of test cultures, although to varying degrees. Gram-negative bacteria were found to be the least sensitive, consistent with numerous previously reported data.

The list of test cultures includes pathogens for both honeybees and humans. It is well known that the most destructive brood disease of honeybees called American foulbrood (AFB), caused by the Gram-positive spore-forming bacterium *Paenibacillus larvae*, is a serious problem for global beekeeping [61]. Therefore, great efforts are being made to discover natural remedies that can replace the currently banned antibiotics and other synthetic drugs that have been widely used for a long time to combat this infection. These natural remedies include propolis, whose anti-AFB activity has been associated with phenols, such as flavonoids, phenolcarboxylic and hydroxycinnamic acids and their esters [35,60,62], as well as triterpenoids [63].

Comparison of the anti-AFB activity of propolis from SLBs that do not contain these components with the previously obtained MIC values [60] shows that it is at a level typical for propolis of the ‘birch’ type (15.6–31.8 μg mL^−1^) in the case of extracts 1 and 4 containing phenolic lipids and triterpenoids. Extracts 2 and 3, with a high content of diterpenoids, are characterized by MIC values approximately two times higher than those of the least potent ‘aspen’ type European propolis (62.4 μg mL^−1^).

Ether extracts of Argentine propolis (1–4) also showed high activity against the tested human pathogens: the Gram-positive bacterium *S. aureus* and the Gram-negative bacteria *E. coli* and *P. aeruginosa*, as well as the fungus *C. albicans*. Particularly sensitive to the action of extracts 2 and 3 rich in diterpenoids were the bacteria *B. cereus* and *B. subtilis*, which can cause food poisoning in humans. Interestingly, the MIC values given in Table 5 for the propolis extract 4 of *T. fiebrigi* in relation to *S. aureus*, *P. aeruginosa* and *C. albicans* turned out to be approximately an order of magnitude lower than in the case of Brazilian propolis of the same species of bees [9], which did not contain phenolic lipids and triterpenoids (Section 3.1).

### 3.5. Anti-Biofilm Action

The high activity of Argentinian propolis from SLBs against certain microorganisms may be due to various reasons, one of which may be the anti-biofilm-forming effect of its extracts [11,64]. In this study, the effect of four propolis extracts on the ability of bacterial cells to form biofilm was determined using the crystal violet staining method. Three Gram-positive strains (*P. larvae*, *B. cereus* and *B. subtilis*) and one Gram-negative strain (*E. coli*) were used for the experiment. The effect of propolis extracts at concentrations below the determined MIC values (Table 5) on biofilm formation is shown in Appendix A. All tested extracts in the range of concentrations from 1/32 MIC to ½ MIC statistically significantly (*p* < 0.05) reduced the biofilm biomass of both Gram-positive and Gram-negative strains relative to control cells. Furthermore, extracts at the concentrations determined did not inhibit bacterial growth statistically significantly (data not shown), but sub-MIC concentrations reduced the ability of these strains to form biofilm. Thus, the action of extracts from SLBs as anti-biofilm agents was not associated with bacterial growth inhibition. *S. jujuyensis* propolis also reduced the formation of *Staphylococcus aureus* and *P. aeruginosa* biofilm in a previous study, and this, according to the authors [11], was also not associated with the inhibition of bacterial growth.

Based on the obtained data, minimum biofilm inhibitory concentration (MBIC_50_) values were determined (Table 6). The MBIC_50_ values of extracts varied between 0.035 and 311.466 µg mL^−1^. The results showed that all extracts inhibit biofilm formation but to varying degrees. The strongest activity for all extracts was observed against *B. cereus* and *B. subtilis* strains. In contrast, they were much less effective against *P. larvae* and *E. coli* strains.

In general, the lowest MBIC_50_ values (below 0.06 µg mL^−1^) against *B. cereus* and *B. subtilis* strains were observed for extracts 2 and 3. Sufficiently strong anti-biofilm activity against the same species was also observed in extracts 1 and 4. The higher MBIC_50_ values were obtained for the Gram-negative *E. coli* strain, but the activity of the extracts was arranged in the same order as for the Gram-positive bacteria *B. cereus* and *B. subtilis*: extracts 3 and 2 significantly more strongly inhibited the formation of biofilm in *E. coli* compared to extracts 1 and 4. On the contrary, extracts 1 and 4 showed stronger inhibition of biofilm formation in *P. larvae*.

Thus, the high anti-biofilm activity of all four types of SLB propolis was demonstrated. The Gram-negative bacterium *E. coli* was the least sensitive, which is consistent with the literature data [64]; however, even against that, the effect of extracts was manifested at concentrations less than 1 mg mL^−1^. Also noteworthy is the fact that in this case, there is a pairwise similarity of the tested propolis samples, which is undoubtedly related to their chemical composition.

## Figures and Tables

**Figure 1 molecules-27-07686-f001:**
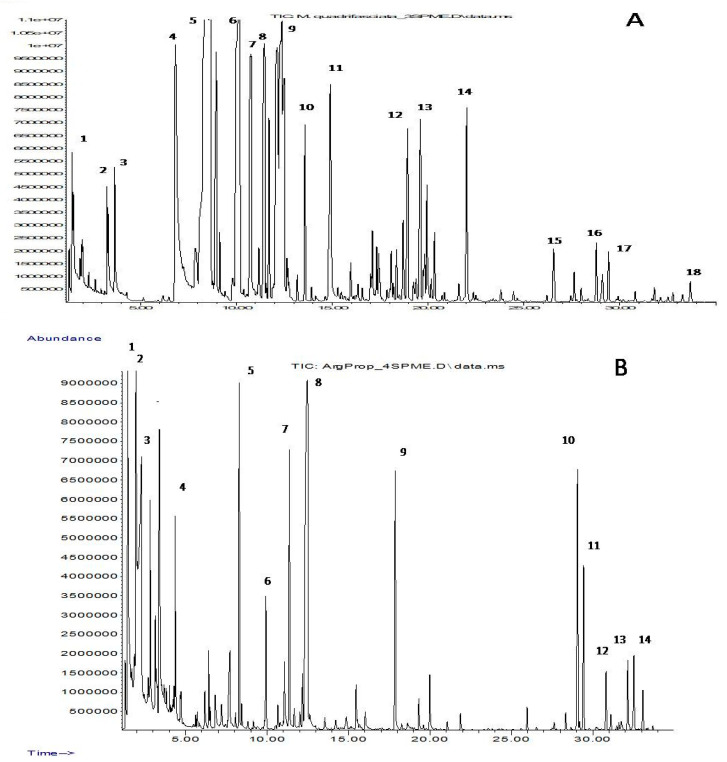
Chromatogram of the volatile components in propolis *M. quadrifasciata* (**A**) and *T. fiebrigi* (**B**). (**A**) 1—ethanol, 2—pyridine, 3—toluene, 4—styrene, 5—α-pinene, 6—β-pinene, 7—myrcene, 8—3-carene, 9—limonene, 10—γ-terpinene, 11—terpinolene, 12—4-terpineol, 13—α-terpineol, 14—methyl carvocrol, 15—α-copaene, 16—longifolene, 17—β-caryophyllene and 18—δ-cadinene. (**B**) 1—ethanol, 2—ethyl acetate, 3—acetic acid, 4—ethyl butanoate, 5—α-pinene, 6—β-pinene, 7—3-carene, 8—2-ethylhexan-1-ol, 9—2-ethylhexyl acetate, 10—α-gurjunene, 11—β-caryophyllene, 12—α-humulene, 13—β-selinene and 14—α-selinene.

**Figure 2 molecules-27-07686-f002:**
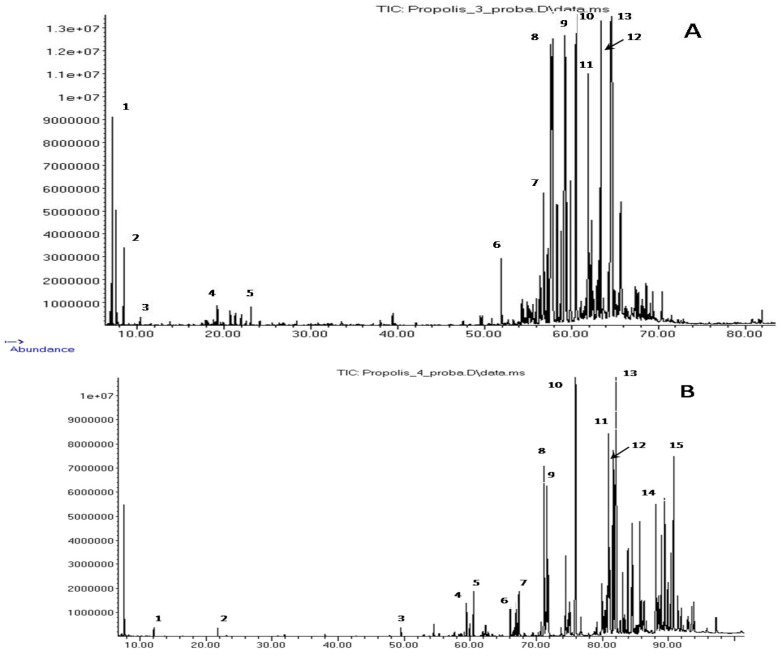
Chromatogram of the extractive components of propolis *M. quadrifasciata* (**A**) and *T. fiebrigi* (**B**). (**A**) 1—α-pinene; 2—β-pinene; 3—limonene; 4—(*E*)-verbenol, mono-TMS; 5—α-terpineol, mono-TMS; 6—manool, mono-TMS; 7—pimaric acid, mono-TMS; 8—(*E*)-communic acid, 9—dehydroabietic acid, 10—13-epi-cupressic acid, di-TMS; 11—unidentified diterpenoid, 12—imbricatoloic acid, di-TMS; and 13—isocupressic acid, di-TMS. (**B**) 1—lactic acid, di-TMS; 2—glycerol, tri-TMS; 3—palmitic acid, mono-TMS; 4—dehydroabietic acid, mono-TMS; 5—13-epi-cupressic acid, di-TMS; 6—3-heptadienylphenol, C17:2, mono-TMS; 7—5-heptadecylresorcinol, di-TMS; 8—5-heptadecadi-8,11-enylresorcinol, di-TMS; 9—5-heptadecenylresorcinol, di-TMS; 10—*n*-hentriacontane; 11—β-amyrin, mono-TMS; 12—α-amyrin, mono-TMS; 13—cycloartenol, mono-TMS; 14—tetratriacontanol, mono-TMS; and 15—mangiferonic acid, di-TMS.

**Figure 3 molecules-27-07686-f003:**
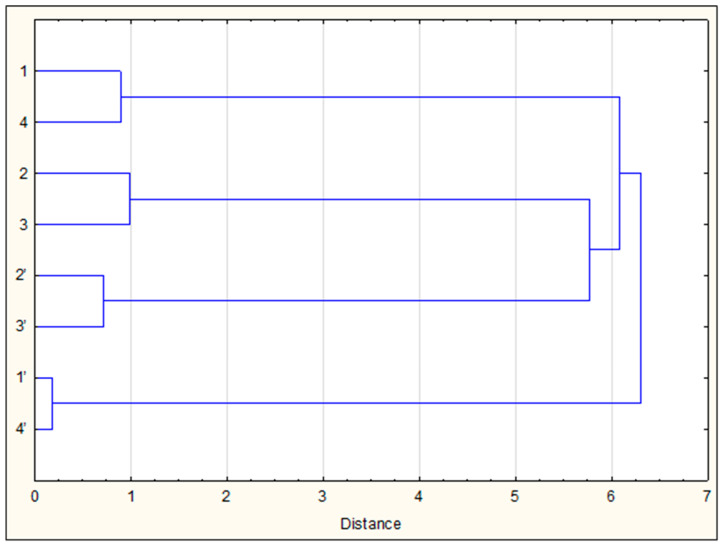
Dendrogram of the chemical similarity of VOCs (1–4) and ether extracts (1’–4’) of the study of SLB propolis samples.

**Table 1 molecules-27-07686-t001:** Scientific and local names of bees whose propolis was studied.

Sample Number	Scientific Name	Common Name
1	*Scaptotrigona* aff. *postica* (Latreille, 1807)	Mandaguarí negra
2	*Tetragona clavipes* (Fabricius, 1804)	Borá
3	*Melipona quadrifasciata quadrifasciata* (le Peletier, 1836)	Mandazaia
4	*Tetragonisca fiebrigi* (Schwarz, 1938)	Yateí

**Table 2 molecules-27-07686-t002:** Group composition of volatile compounds in Argentine propolis of stingless bees: 1—Scaptotrigona postica, 2—Tetragona clavipes, 3—Melipona quadrifasciata quadrifasciata and 4—Tetragonisca fiebrigi.

Group of Compounds	Relative Composition (% of TIC)			
	1	2	3	4
Monoterpene hydrocarbons, including:	6.01 **(5) ***	41.14 **(18)**	59.50 **(16)**	13.96 **(9)**
- Bornylene	- **	-	1.24	-
- Tricyclene	-	0.81	1.29	-
- α-Thujene	-	-	-	0.20
- α-Pinene	1.98	12.16	20.97	5.34
- Camphene	0.08	2.63	3.29	-
- β-Pinene	0.52	8.57	10.10	1.96
- Myrcene	-	2.19	4.29	0.40
- 3-Carene	3.17	2.35	3.93	4.57
- β-Phellandrene	-	1.77	-	-
- Limonene	0.25	1.77	6.74	0.81
- γ-Terpinene	-	1.37	1.37	0.25
- Terpinolene	-	3.18	3.55	0.32
Monoterpenoids, including:	0.40 **(3)**	10.35 **(23)**	10.60 **(27)**	0.19 **(1)**
- *trans*-Sabinene hydrate	-	0.18	0.10	-
- *cis*-Sabinene hydrate	-	-	0.17	-
- *trans*-Linalool oxide	0.16	-	-	0.19
- Linalool	-	0.26	0.11	-
- Fenchol	-	0.45	0.36	-
- *trans*-Pinocarveol	-	0.82	-	-
- Camphor	-	0.35	0.47	-
- Borneol	-	0.72	0.45	-
- 4-Terpineol	-	2.10	1.83	-
- α-Terpineol	-	0.66	2.19	-
- Bornyl acetate	-	0.67	0.08	-
**Sesquiterpene hydrocarbons, including:**	**5.98 (10)**	**14.71 (23)**	**2.91 (14)**	**12.25 (10)**
- α-Cubebene	0.10	1.30	0.63	-
- α-Copaene	0.31	0.73	0.24	trace ***
- β-Bourbonene	-	0.42	0.12	-
- α-Gurjunene	2.55	-	-	5.14
- Longifolene	-	0.37	0.54	-
- β-Funebrene	-	0.34	0.34	-
- β-Caryophyllene	1.41	3.02	0.45	2.83
- α-Humulene	0.40	0.83	0.08	0.98
- Alloaromedendrene	0.16	-	-	-
- γ-Muurolene	-	0.91	0.09	-
- Germacrene D	-	1.49	-	-
- β-Selinene	0.47	-	-	1.28
- α-Selinene	0.43	-	-	1.41
- γ-Cadinene	-	0.72	0.08	-
- δ-Cadinene	0.09	1.19	0.23	trace
**Sesquiterpenoids, including:**	**-**	**0.81 (9)**	**0.08 (1)**	**-**
- Spathulenol	-	0.13	-	-
- Caryophyllene oxide	-	0.10	-	-
- Humulene II epoxide	-	0.02	-	-
- Junenol	-	0.24	-	-
- α-Cedrol	-	0.08	0.08	-
- α-Cadinol	-	0.07	-	-
**Aliphatic alcohols, including:**	**33.40 (5)**	**3.83 (4)**	**4.74 (4)**	**29.39 (9)**
- Ethanol	3.36	1.70	1.06	8.35
- 1-Propanol	1.95	0.97	-	1.22
- 1-Butanol	-	-	0.08	-
- Isopentanol	-	1.10	0.03	1.05
- 1-Hexanol	0.43	-	-	0.63
- 2-Heptanol	0.47	-	-	0.18
- 2-Ethylhexan-1-ol	27.04	-	3.58	17.14
- 1-Octanol	-	0.06	-	
- 1-Nonanol	0.02	-	-	0.12
**Aliphatic acids, including:**	**8.70 (2)**	**4.44 (3)**	**0.25 (1)**	**12.48 (2)**
- Acetic acid	7.60	2.68	0.25	12.24
- Propanoic acid	1.10	1.33	-	0.24
- Butyric acid	-	0.43	-	-
**Esters, including:**	**30.20 (18)**	**14.54 (17)**	**0.51 (2)**	**23.26 (17)**
- Ethyl acetate	4.30	5.31	0.41	6.83
- Methyl propionate	1.86	trace	-	-
- Ethyl propionate	6.60	2.87	-	2.76
- Ethyl butanoate	-	-	-	2.63
- Isopentyl formate	1.98	-	-	-
- Ethyl lactate	-	-	-	0.67
- Propyl propionate	5.14	1.90	-	-
- Butyl propionate	0.24	-	-	-
- Propyl butanoate	-	0.89	-	-
- Isoamyl propionate	0.48	-	-	-
- Ethyl hexanoate	0.32	-	-	1.26
- 2-Ethylhexyl acetate	12.58	0.22	0.10	0.26
- Ethyl octanoate	0.07	0.93	-	0.94
**Aromatics, including:**	**2.01 (5)**	**8.78 (4)**	**17.50 (12)**	**2.40 (5)**
- Toluene	-	-	1.07	0.80
- Styrene	1.65	1.39	6.46	0.68
- Phenol	0.17	-	-	-
- Benzaldehyde	-	-	0.16	-
- *p*-Cymene	0.09	6.85	5.70	0.28
- Benzyl alcohol	-	-	0.24	0.32
- 2-Phenyl ethanol	0.08	-	-	0.32
- Benzoic acid	0.02	-	-	-
- *p*-Cymen-8-ol	-	0.28	0.26	-
- Methyl chavicol (estragol)	-	-	1.14	-
- Thymol methyl ether	-	0.25	0.18	-
**Other, including:**	**0.27 (2)**	**2.70 (1)**	**1.90 (4)**	**3.08 (3)**
- Acetaldehyde	-	-	0.06	-
- Pyridine	-	2.70	0.73	-
- γ-Butyrolactone	-	-	-	1.99
- γ-Valerolactone	0.10	-	-	0.09
**NN**	**0.52 (5)**	**0.57 (2)**	**1.97 (6)**	**2.90 (6)**

*—not detected; ** the number of peaks in a particular group of components is given in parentheses; *** trace—below 0.01% of TIC.

**Table 3 molecules-27-07686-t003:** Group composition of diethyl ether extracts of Argentine propolis of SLBs (1—*S. postica*; 2—*T. clavipes*; 3—*M. quadrifasciata quadrifasciata*; 4—*T. fiebrigi*).

Group of Compounds	Relative Composition (% of TIC)			
	1	2	3	4
**Monoterpenoids, including:**	**- ***	**1.10 (11) ****	**4.09 (20)**	**-**
- α-Pinene	-	0.86	1.89	-
- β-Pinene	-	0.29	0.73	-
- Camphene hydrate	-	0.06	0.16	-
**Sesquiterpenoids, including:**	**trace *** (5)**	**2.06 (20)**	**0.23 (3)**	**-**
- β-Caryophyllene	trace	0.32	-	-
- Germacrene D	-	0.36	-	-
- Bicyclogermacrene	-	0.11	-	-
- α-Cadinol, TMS	-	0.25	-	-
**Diterpene acids, including:**	**0.03 (4)**	**57.52 (16)**	**64.81 (16)**	**2.38 (9)**
- Pimaric acid, TMS	-	1.34	2.20	trace
- Isopimaric acid, TMS	trace	4.11	5.79	-
- Communic acid, TMS	-	5.32	5.90	-
- Abietic acid, TMS	trace	6.43	2.23	0.13
- Dehydroabietic acid, TMS	0.01	3.10	8.10	0.08
- Neoabietic acid, TMS	-	7.03	1.39	0.07
- 13-*epi*-Cupresic acid, TMS	trace	5.75	10.35	-
- Isocupresic acid, di-TMS	-	6.67	15.20	-
- Imbricatoloic acid, di-TMS	-	4.08	8.48	-
**Other diterpenoids, including:**	**-**	**6.61 (16)**	**20.36 (22)**	**0.62 (2)**
- Sandaracopimarinal	-	0.06	0.09	-
- 13-*epi*-Manool, TMS	-	0.22	0.94	-
- Totarol, TMS	-	2.83	5.99	trace
- Ferruginol, TMS	-	0.90	0.45	-
- Copalool, TMS	-	0.15	0.79	-
- Unidentified diterpenoid, RI 2497	-	0.65	4.81	-
**Triterpenoids, including:**	**76.93 (29)**	**12.54 (22)**	**0.39 (3)**	**49.91 (32)**
- Cycloartenol, TMS	7.50	-	-	11.54
- Lupeol, TMS	9.87	-	-	3.444
- α-Amyrin, TMS	9.91	0.40	0.07	3.96
- β-Amyrin, TMS	8.73	0.69	0.11	5.08
- Dipterocarpol, TMS	-	-	-	1.87
- Unidentified triterpenoid, RI 3417	7.50	-	-	-
- Unidentified triterpenoid, RI 3645	2.53	-	-	0.71
- Isomangiferolic acid, di-TMS	2.41	-	-	2.06
- Mangiferolic acid, di-TMS	2.46	-	-	1.99
- Mangiferonic acid, TMS	3.81	-	-	5.01
**Phenols, including:**	**1.60 (7)**	**-**	**-**	**1.03 (6)**
- Hydroginkgol, TMS	0.07	-	-	0.08
- 3-Heptadeca-9,12-dienylphenol, TMS	0.30	-	-	0.55
- 3-Heptadecenylphenol, isomer 1, TMS	0.07	-	-	0.10
- 3-Heptadecenylphenol, isomer 2, TMS	0.15	-	-	0.15
- 3-Heptadecylphenol, TMS	0.12	-	-	0.14
- 3-Heptadecyl-13-hydroxyphenol, di-TMS	0.84	-	-	-
**Resorcinols, including:**	**8.16 (10)**	**-**	**-**	**9.38 (9)**
- 5-Pentadecenyl resorcinol, di-TMS	0.61	-	-	0.23
- 5-Pentadecyl resorcinol, di-TMS	1.00	-	-	0.61
- 5-(8,11-Heptadecadienyl) resorcinol, di-TMS	3.04	-	-	4.29
- 5-Heptadecatrienyl resorcinol, di-TMS	-	-	-	0.25
- 5-Heptadecenyl resorcinol, isomer 1, di-TMS	0.51	-	-	0.25
- 5-Heptadecenyl resorcinol, isomer 2, di-TMS	0.60	-	-	0.81
- 5-Heptadecenyl resorcinol, isomer 3, di-TMS	2.32	-	-	1.40
- 5-Heptadecyl resorcinol, di-TMS	-	-	-	1.14
- 5-Nonadecenyl resorcinol, di-TMS	0.59	-	-	-
**Salicylates, including:**	**2.76 (6)**	**-**	**-**	**4.07 (8)**
- 6-Pentadecenyl salicylic acid, di-TMS	0.06	-	-	-
- 6-Heptadecadienyl salicylic acid, di-TMS	0.82	-	-	1.63
- 6-Heptadecenyl salicylic acid, isomer 1, di-TMS	0.25	-	-	0.09
- 6-Heptadecenyl salicylic acid, isomer 2, di-TMS	0.21	-	-	0.39
- 6-Heptadecenyl salicylic acid, isomer 3, di-TMS	0.97	-	-	0.61
- 6-Heptadecyl salicylic acid, di-TMS	-	-	-	0.61
- 6-(12-Hydroxyheptadecyl) salicylic acid, tri-TMS	0.45	-	-	-
**Aliphatic alcohols, including:**	**0.04 (1)**	**3.32 (4)**	**-**	**6.73 (7)**
- Octacosanol, TMS	-	0.22	-	-
- Triacontanol, TMS	-	1.69	-	-
- Dotriacontanol, TMS	-	1.22	-	2.68
- Tetratriacontanol, TMS	-	0.19	-	3.72
**Aliphatic acids, including:**	**0.39 (10)**	**0.65 (7)**	**-**	**1.15 (10)**
- Palmitic acid, TMS	0.12	0.15	-	0.21
- Linoleic acid, TMS	0.04	trace	-	0.06
- Oleic acid, TMS	0.12	0.21	-	0.22
**Aliphatic esters**	**2.14 (2)**	**4.57 (7)**	**-**	**4.56 (6)**
- Triacontyl acetate	-	1.44		-
- Dotriacontyl acetate	-	1.38		-
- Tetratriacontyl acetate	-	0.30		3.15
- Unknown aliphaic acids acetate (2 isomers)	2.14	-		0.51
**Other**	**0.61 (9)**	**2.56 (5)**	**1.05 (1)**	**11.68 (9)**
**NN**	**7.44 (8)**	**9.48 (31)**	**9.08 (21)**	**8.49 (17)**

* —not detected; ** the number of peaks in a particular group of components is given in parentheses; *** trace—below 0.01% of TIC.

**Table 4 molecules-27-07686-t004:** IC_50_ values of propolis extracts 1–4 for fibroblasts and selected cancer cell lines after 24 and 48 h of treatment.

Propolis Sample	IC_50_ (μg mL^−1^)											
	*A375*		*C32*		SCC-25		AGS		DLD-1		Fibroblasts	
	24 h	48 h	24 h	48 h	24 h	48 h	24 h	48 h	24 h	48 h	24 h	48 h
1	52.4 ± 3.9	29.4 ± 2.1	34.0 ± 1.8	31.4 ± 2.2	26.3 ± 1.4	21.6 ± 1.6	54.9 ± 3.6	29.3 ± 1.5	67.3 ± 4.2	31.9 ± 2.0	35.0 ± 2.1	33.9 ± 1.9
2	79.4 ± 6.1	16.8 ± 1.5	37.7 ± 2.3	17.3 ± 1.3	22.3 ± 1.6	20.5 ± 1.2	38.2 ± 2.4	12.6 ± 0.9	50.4 ± 2.3	20.2 ± 1.5	52.0 ± 4.6	34.2 ± 1.4
3	89.6 ± 4.2	43.9 ± 3.6	40.5 ± 2.8	34.1 ± 2.4	42.9 ± 3.5	33.4 ± 1.8	51.1 ± 2.9	23.0 ± 1.4	70.9 ± 5.9	32.9 ± 2.4	68.9 ± 4.2	57.9 ± 3.7
4	54.6 ± 3.1	36.1 ± 2.4	45.0 ± 3.1	43.0 ± 3.0	29.2 ± 2.3	26.5 ± 3.0	55.6 ± 4.6	43.0 ± 3.1	70.4 ± 5.1	44.3 ± 3.6	53.8 ± 5.0	43.2 ± 2.9

**Table 5 molecules-27-07686-t005:** Antimicrobial activity (MIC and MBC/MFC) of extracts from stingless bees and honeybees propolis (1—*S. postica*, 2—*T. clavipes,* 3—*M. quadrifasciata quadrifasciata,* 4—*T. fiebrigi*; 5—*A. mellifera*).

Sample	Gram-Positive Bacteria				Gram-Negative Bacteria		Fungus
	*P. larvae*	*S. aureus*	*B. cereus*	*B. subtilis*	*E. coli*	*P. aeruginosa*	*C. albicans*
MIC, µg mL^−1^ (reading after 48 h)							
1	31.25	125	1.95	7.81	500	500	125
2	125	31.25	0.12	0.12	125	500	31.25
3	125	31.25	0.12	0.12	125	500	31.25
4	31.25	31.25	0.49	0.49	500	500	31.25
5	7.8–62.4 ^a^	16–62 ^b^	31–62 ^b^	-	>2500 ^b^	250–500 ^b^	39–312 ^a^
MBC/MFC, µg mL^−1^ (reading after 48 h)							
1	125	2000	7.81	31.25	2000	>2000	500
2	500	125	0.49	0.49	500	>2000	500
3	500	125	0.49	0.49	500	>2000	125
4	125	2000	1.95	1.95	2000	>2000	500

^a^—Isidorov et al. [60], 9 samples; ^b^—Isidorov et al. [35], 6 samples.

**Table 6 molecules-27-07686-t006:** Minimmum biofilm inhibitory concentration (MBIC_50_ µg mL^−1^) of extracts from stingless bees (1—*S. postica*, 2—*T. clavipes*, 3—*M. q. quadrifasciata*, 4—*T. fiebrigi*).

Sample	Gram-Positive Bacteria			Gram-Negative Bacteria
	*P. larvae* ATCC 9545	*B. cereus* ATCC 10987	*B. subtilis* ATCC 6633	*E. coli* ATCC 11229
1	23.57 ± 2.53	1.45 ± 0.37	5.90 ± 0.90	280.86 ± 57.60 *
2	111.97 ± 25.51	0.06 ± 0.01	0.049 ± 0.001	53.39 ± 4.60
3	97.91 ± 4.06	0.04 ± 0.002	0.035 ± 0.001	50.76 ± 2.97
4	16.57 ± 2.00	0.21 ± 0.02	0.21 ± 0.011	311.47 ± 29.82

MBIC_50_: minimum biofilm inhibitory concentration, defined as the lowest concentration of extract capable of reducing biofilm formation by 50% compared to the control assay (* IC_25_**,** µg mL^−1^**).**

## Data Availability

The study did not report any data.

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
