# Peer review of "Chemical Composition and Biological Activity of Argentinian Propolis of Four Species of Stingless Bees"

_molecules, 2022, doi:10.3390/molecules27227686_

Round 1
Reviewer 1 Report
The current manuscript is focused on chemical characterization and biological (antimicrobial and anticancer) activity of Argentinian propolis from four stingless bee species. A number of compounds are found and identified by GC-MS in both volatile fractions and diethyl ether extracts. Based on the chemistry, suggestions for the plant sources of the samples are made. The manuscript is of interest and adds knowledge in the chemistry of propolis from different stingless bee species. The data are clearly presented and deserve publication after minor revision, as follows:
1. In the title of Figure 1 several compounds are mentioned for two of the propolis samples. It would be better if authors mark the major compounds on the presented chromatograms.
2. The GC chromatograms of the diethyl ether extracts of the selected samples is also good to be inserted in the manuscript.
Author Response
Reply to comment 1. Unfortunately, when loading the figure into the text of the article, the numbering and corrections on the chromatogram disappeared. This has been corrected in the new edition.
Reply to comment 2. In accordance with the recommendation of the reviewer, the chromatograms of ether extracts of selected propolis samples are included in the manuscript (Fig. 2 in the new edition of the article).
Reviewer 2 Report
Dear Authors
The MS entitled “Chemical composition and biological activity of Argentinian 2 propolis of four species of stingless bees” was thoroughly reviewed. The article is technically correct however, there are some serious concerns in the formulation of MS.
General comments:
· Provide the Email addresses of all the authors in authors affiliation sections (according to the molecules format).
Abstract:
· Spacing between line 14 and 15 should be corrected.
· The abstract must be precise.
Introduction:
· References citations should be in numerical [1] and in order (starting from 1) as the format suggests.
· Page 2, line 50-52: Correct the grammar.
· Page 2 line 94: Correctly write the full name of abbreviation (HS-SPME/GC-MS).
· Page 4 and page 5: check font sizes.
Results and discussion
· Modify table 1. The row having these numbers 6.01 (5)* 41.14 (18) 59.50 (16) 13.96 (9) are confusing. Also, what 5* means in second row?
· Clear Figures should be presented.
· Why no reference drug was used in MBIC50 measurements?
References
· Also, the references should be revised and modified according to the journal format.
Author Response
Note 1. Provide the Email addresses of all the authors in authors affiliation sections (according to the molecules format).
Reply to note: This section contains the email address of the corresponding author. Among the articles already published in "Molecules" we have not come across a single one in which the e-mail addresses of all co-authors would be given.
Note 2. Spacing between line 14 and 15 should be corrected
Reply to note: Fix done.
Note 3. The abstract must be precise
Reply to note: It is not clear to us what is meant, however, a phrase has been added to the abstract regarding the anti-cancer activity of propolis.
Note 4. References citations should be in numerical [1] and in order (starting from 1) as the format suggests.
Reply to note: Citation in text changed to digital.
Note 5. Page 2, line 50-52: Correct the grammar.
Reply to note. This has been changed.
Note 6. Page 2 line 94: Correctly write the full name of abbreviation (HS-SPME/GC-MS).
Reply. Full name of abbreviation given
Note 7. Page 4 and page 5: check font sizes.
Reply. Font size checked
Note 8. Modify table 1. The row having these numbers 6.01 (5)* 41.14 (18) 59.50 (16) 13.96 (9) are confusing. Also, what 5* means in second row?
Reply. This is a clear misunderstanding: apparently the reviewer did not pay attention to the explanations given under the table, marked with asterisks. In our opinion, the table does not need to be modified.
Note 9. Clear Figures should be presented
Reply. We tried to give clear drawings of chromatograms and
dendrogram.
Note 10. Why no reference drug was used in MBIC50 measurements.
Reply. Unfortunately, we are not aware of any drug that could serve as a reference in the study of the antibiofilm activity of natural agents. Ciprofloxacin is named as such in some papers (see ref. [11]), but it was found to be inactive against Staphylococcus aureus, as well as against E. coli (see Antibiotics, 2021 Sep 24;10(10):1159, as well as Indian J Med Res 141, March 2015, pp 343-353). In the available literature, it was not possible to find information on a medical product that exhibits antibiofilm activity in the case of Paenibacillus larvae.
Note 11. Also, the references should be revised and modified according to the journal format.
Reply. The list of references has been revised and changed.
Round 2
Reviewer 2 Report
Dear Authors
I have gone through the revised version.
Usually, the templet of this journal requires all the authors email addresses with their name’s abbreviations. (see the templet carefully). If the editor has no issue with this then it is ok.
secondly, the author should know the meaning of words such as precise, conclusive, short, brief, explanatory, etc. he precision of abstract was meant to highlight the most potent or significant effect of tested sample against the cancer cell lines with exact values and also in the case of antibiofilm activities. However, the authors failed to state the exact values. As far as tables are concerned, the numbers in small brackets are confusing. As it shows the number of peaks in the GC or identified compounds number?
The rest of MS is OK.